# A computational text analysis investigation of the relation between personal and linguistic agency

Almog Simchon [1,2 ✉], Britt Hadar [3] & Michael Gilead [2,4 ✉]

Previous psycholinguistic findings showed that linguistic framing – such as the use of passive voice - influences the level of agency attributed to other people. To investigate whether passive voice use relates to people's personal sense of agency, we conducted three studies in which we analyzed existing experimental and observational data. In Study 1 (N = 835) we show that sense of personal agency, operationalized between participants as recalling instances of having more or less power over others, affects the use of agentive language. In Study 2 (N = 2.7 M) we show that increased personal agency (operationalized as one's social media followership) is associated with more agentive language. In Study 3 and its two replications (N = 43,140) we demonstrate using Reddit data that the language of individuals who post on the r/depression subreddit is less agentive. Together, these findings advance our understanding of the nuanced relationship between personal and linguistic agency.

[1] School of Psychological Science, University of Bristol, 12A Priory Road, Bristol BS8 1TU, UK. [2] Department of Psychology, Ben-Gurion University of the Negev, Israel, POB 653 Beer Sheva 84105, Israel. [3] Baruch Ivcher School of Psychology, Reichman University, Israel, 8 Ha'universita St, POB 167 Herzliya 46150, Israel. [4] The School of Psychological Sciences, Tel Aviv University, Israel, POB 39040 Tel Aviv 6997801, Israel. ✉email: almogsi@post.bgu.ac.il; michaelgil@tauex.tau.ac.il

The relation between language and thought has long fascinated both scholars and artists. One of the most prominent voices in this discussion was George Orwell, whose novel 1984[1] famously depicts a dystopia in which a totalitarian regime controls the public by mandating a language designed to restrict thought. Orwell further explored this theme in his essay "Politics and the English Language"[2] wherein he proposed that certain linguistic structures such as the use of the passive voice may facilitate oppressive ideologies: passive sentences (e.g., "Her arms were clasped tightly around Winston") and active sentences (e.g., "She clasped her arms tightly around Winston") supposedly describe the same activity, however, whereas the active voice highlights the subject of the sentence, the passive voice diminishes or eliminates it[3,4]. Orwell argued that the use of non-agentive language undercuts the self-agency ascribed to the individual, and may be used to disempower people. Interestingly, despite his abovementioned objection to the passive voice, he revised earlier versions of the novel 1984 to include numerous passive constructions—supposedly to convey the feeling of a world wherein individuals have no control over their lives[5].

Aside from being an effective literary device, social thinkers have long suggested that the use of passive language, and more generally, non-agentive language, is strongly related to the degree to which people possess personal agency[6–10]. Personal agency is the ability of humans to exert control over the world and the self, and is manifested in (i) the degree to which people are in control of their actions; (ii) the degree that they are in control of outcomes and resources (e.g., sense of social power[11]); and (iii) the degree to which people believe they possess (i) and (ii)--namely, people's subjective sense of control[12]. Control has long been considered a fundamental need (i.e., "der Wille zur Macht", translated as the "will to power" in the philosophy of Nietzsche[13]) and much research suggests that it is crucial for people's well-being[12,14–18].

Previous psycholinguistic findings showed that linguistic framing influences the level of agency attributed to other people. For example, describing incidents in an agentive language led to greater attribution of blame by observers, and resulted in harsher punishments than in non-agentive language[19]. Moreover, in languages that highlight the agent in accidental incidents (e.g., English), the agents are better encoded in memory than in languages that do not (e.g., Japanese)[20]. Such findings concerning the effects of agentive language join a rich body of work concerning linguistic (e.g., linguistic abstractness[21]) and paralinguistic (e.g., voice pitch[22]) cues that affect our attributions of personal agency to a third party. However, the philosophical claim concerning the importance of linguistic agency goes beyond the impact of agency attributes on other people but also includes its influence on the speaker's own sense of psychological agency

Much of the previous work concerning the relation between linguistic and personal agency relied on qualitative discourse analyses. For example, a qualitative report suggests that individuals dealing with chronic pain often discuss their struggles using passive voice, supposedly reflecting a sense of reduced personal agency[23]. Furthermore, qualitative descriptions of people's reconstructions of psychological therapy show that patients describe periods of psychological hardship in a passive voice and that they often use more agentive language when describing the process of improvement[24].

Two studies that involved quantitative analyses examined the use of passive language in the context of emotional hardships and its potential link to a diminished sense of personal agency. One has shown that individuals with OCD tend to use less agentive language[25]. The second demonstrated that skin-conductance levels tend to increase when individuals describe traumatic experiences in a passive voice[26]. This finding suggests that passive language was associated with experiences of high negative arousal, during the recollection of events that are often characterized by deprivation of control.

Aside from analyses of personal agency and agentive language in the context of psychopathology, research has also examined social power, which is another correlate of personal agency, related to agentive language. One notable example comes from Duranti[27], who studied the local language of a western Samoan village, a language that can explicitly mark an agent in its grammar. He discovered that agentive language use corresponded with the speaker's social position. That is speakers with a higher status within the village hierarchy (thereby having higher control over the community's decisions) tended to use more agentive language. While such anthropological studies are clearly informative and compelling, there is also a need for quantitative analyses that systematically examine the link between linguistic agency and personal control—and their extent and prevalence.

In light of this, in the current research, we sought to understand whether personal agency is reflected in the extent to which individuals use agentive language. Specifically, we aimed to explore whether various factors (social power, social rank, and participation in a depression forum) characterized by personal agency are reflected in the extent to which individuals use the passive voice.

In Study 1, we examined how participants' lack of sense of agency (i.e., little power over one's actions vs. high power) is reflected in passive language use: the sense of personal agency is dependent on having the power to pursue one's goals effectively. In light of this, a first, straightforward prediction is that when individuals lose the power to affect their surroundings, this will be reflected in diminished linguistic agentivity. Therefore, in Study 1, we examined the effect of a well-established experimental manipulation of a sense of power[28–32] on participants' use of passive voice.

In Study 2 we examined the relationship between an individual's place within a social network and their tendency to use passive language. We primarily focused on their status and power, represented by the number of followers they have, which are fundamental parts of social hierarchy[33–36]. Power and status allow individuals at the top of the hierarchy to have better access to resources, such as money, food, and potential partners, as well as the ability to make decisions for themselves and others[37,38]. Consequently, those with high social rank experience greater control over their own outcomes and the outcomes of others, leading to increased personal agency[39,40] (see also the agentic-communal model of power[41]).

According to influential theories in anthropology and psychology, the number of followers is indicative of one's social rank in that group[42,43]. More followers indicate higher status. As mentioned in the introduction, some initial evidence for a link between social power and agentive language use comes from ethnographic research[27], which discovered that agentive language use corresponded with the speaker's social position. That is, speakers with a higher status within the village hierarchy tended to use more agentive language. Here, we generalize the findings from the western Samoan village to a population of millions and ask whether social ranking in a naturally occurring social network relates to agentive language use. We utilized a large dataset from Twitter (26.4 M messages) to examine this relationship. We hypothesized that the number of followers on social media would be negatively associated with the use of passive voice.

In Study 3, we examined whether the language in a forum designated to the topic of depression is more passive, as would be predicted as a result of the loss of agency experienced by many people with depression. Depression is a debilitating mental illness characterized by recurring episodes of low mood, anhedonia, low self-esteem, and hopelessness (for an exhaustive list see DSM-V[44]). According to the Learned Helplessness Model of

Depression[45], depression arises when a person forms the belief that they have no control over the negative outcomes in their lives. Indeed, previous research has shown that individuals experiencing depression report having a lower sense of efficacy[46], lower sense of control[12], and enhanced external locus of control[47]. People who are experiencing depression often seek solace in online communities wherein they find support and empathy. In light of this, in Study 3, we examined whether people who post in a forum dedicated to depression also express less agentivity in their language. We utilized large datasets from the community network Reddit to test the pre-registered hypothesis that the online communities of people experiencing depression would exhibit more passive voice in their messages than a random sample of other popular communities.

## Methods

**Study 1: does manipulating personal agency affect linguistic agency?** The data analyzed in Study 1 are the same data as in Kasprzyk, L., & Calin-Jageman, R[48]. accessed from https://osf.io/wch5r/. We performed a re-analysis of data from a study that attempted to replicate the findings of another study[28]. In the replication attempt, power was manipulated by using an essay writing task. The data repository contained samples collected online from MTurk and Prolific. Participants were assigned to one of two conditions: high power or low power and were asked to write about incidents where they had power over others (high power) vs. incidents where other people had power over them (low power).

In the mTurk study, 469 responses were collected (182 men; 269 women; 56 did not respond). The average age in the sample was 34.91 (SD = 12.46). Ten observations had missing text and were therefore excluded from the analysis. In the Prolific study, there were 416 collected responses (151 men; 184 women; 81 did not respond). Due to technical problems, only 331 participants had a valid age response. The average age was 27.88 (SD = 9.91). Forty observations had missing text and were therefore excluded from the analysis. The study was conducted under the Departmental IRB guidelines and was ruled "exempt".

In Study 1, as well as in subsequent studies, we quantified non-agentive language as the use of passive voice, which serves as our central measure. Previous studies estimate that 70% of passive voice use comes in the agentless form[49,50], serving as an adequate measure for non-agentive language. To measure passive language in the dataset, we used spaCy[51], a popular and powerful natural language processing tool that, among other features, allows users to tag text containing passive voice in a context-dependent manner (https://spacy.io). The passive voice measure was calculated using the 'spaCyr' package[52] as the number of passive voice auxiliary verbs in each text (see supplementary code in the OSF repository). These include both passives (e.g., "the ball was kicked") and *got* passives (e.g., "the ball got kicked").

To validate that computational extraction of passive voice is indeed associated with non-agentive language, an independent rater coded 100 texts from this sample. The rater was presented with various texts, and their objective was to identify and count instances of non-agentive language within those texts. It was explained to the rater that non-agentive language refers to sentences where the action is described without specifying the person or entity responsible for it. On the other hand, agentive language includes sentences that clearly state the agent or individual who performed the action. In order to determine the number of non-agentive instances in each text, the rater needed to analyze the sentences and identify those that lack a specified agent. Prior to carrying out the rating procedure, the rater was provided with sufficient examples (e.g., The vase broke vs. John broke the vase; The book was put on the table vs. Michelle put the book on the table; The curtain caught fire vs. I set the curtain on fire). For the coding manual, see Supplementary Note 1 in SI.

Here we report Spearman's rank correlation coefficient of the rater's coding and the occurrences of passive voice derived from spaCy, $r = 0.70$, $p < 0.001$. We consider these results as evidence supporting the use of computationally derived passive voice as a marker of non-agentive language in the current sample.

Another manifestation of agentive language is the use of self-referential language. The perception of self-agency entails that the self is the causer of events (e.g., "I call the shots"); as such, an additional, potentially important, dimension of agentive language is how frequently individuals refer to themselves in their narratives[24]. While such self-referential language may be a marker of self-agency, much previous research has shown that self-referential language is increased during depressive episodes, supposedly due to the increased self-focus that is common in depression[53]. Because depression is related to reduced self-agency, it is also possible that self-referential processing will actually be a correlate of reduced self-agency. Given these competing possibilities, we did not have a directional hypothesis concerning the effect of self-referential language and included it in our analysis for exploratory purposes. To measure self-referential language, we used the Linguistic Inquiry and Word Count 2015 (LIWC[54]) I-dictionary and counted the instances of self-referring language.

The specific linguistic features described above capture the degree to which individuals represent a given state using agentive or non-agentive language. These linguistic features are (at least theoretically) independent of the specific content individuals choose to convey. People may describe a high personal agency situation in non-agentive language (e.g., "I was chosen as most likely to succeed") or describe a low agency situation in agentive language (e.g., "I now realize I am worthless"). To automatically measures whether individuals generate content reflecting a sense of *personal* agency, we relied on an approach termed Contextualized Construct Representation (CCR)[55]. CCR is an approach that combines psychological insights with natural language processing techniques. This method leverages large contextual language models like BERT[56] to embed both a validated questionnaire measuring a specific construct of interest and the input text (e.g., social media posts) into a latent semantic space. By calculating the cosine similarity between the construct of interest and the input text, CCR provides a measure of association (i.e., CCR loading). Recent research indicates that CCR outperforms other theory-driven text analysis methods, such as word-counting and word embeddings[55], making it well-suited for our theoretically driven investigation.

In the current study, we focused on three interconnected constructs: sense of control[12], locus of control[57], and depression (measured using the CESD scale[58]). To extract embeddings, we utilized a variant of the SBERT model[59] called 'all-MiniLM-L6-v2', implemented through the "text" r package[60]. We extracted the top embedding layer (layer 6) to generate embedded vectors, resulting in each text being represented by a vector with 384 latent dimensions. To ensure consistency, any negatively loaded items within each construct were rephrased to incorporate negations (e.g., transforming "I felt hopeful about the future" to "I did not feel hopeful about the future"). Subsequently, the items were averaged to create a unified representation of each construct. Because the sense of control scale has more items that reflect the constrained factor than the mastery factor, we only use the constrained items. Hence, high CCR loadings reflect a constrained sense of control. For the locus of control construct, where there were an equal number of items for the internal and external factors, we constructed a single anchored vector that represents a continuum of internal-external locus of control based

on the 23 non-filler items[61]; therefore high CCR loading reflects greater internal vs. external locus of control.

**Study 2: social rank and passive voice.** We used the Symbolic Cognition and Interaction lab twitter database[62]. Tweets were collected via Twitter's dedicated API from across the United States, including all 50 states and the District of Columbia. We extracted tweets between April 2019 and June 2019. The study was conducted under the Departmental IRB guidelines and was ruled "exempt".

Our sample size consisted of 26,473,715 tweets, all were in the English language, and all were original (i.e., retweets were filtered out). Text cleaning included the removal of links, tags, and emoticons before any linguistic analysis. Passive voice was extracted the same way as in Study 1.

Same as in Study 1, here again, the independent rater coded 100 texts from a different sample of the same population, and counted instances of non-agentive language. The Spearman's rank correlation coefficient of the rater's coding and the occurrences of passive voice derived from spaCy is positive and sufficient to be a marker of non-agentive language in tweets, $r = 0.54$, $p < 0.001$.

**Study 3: posting on r/depression subreddit and passive voice.** In Study 3a, we collected 10,000 messages from the depression (https://www.reddit.com/r/depression) subreddit (i.e., Reddit community), and 100 messages from 100 randomly selected subreddits (sampled from a list of 1000 popular subreddit, see Supplementary Note 2 in SI). All data were collected via Pushshift API[63] The messages ranged in their time between November 2019 and July 2020. The study was conducted under the Departmental IRB guidelines and was ruled "exempt".

Our sample size consisted of 10,000 messages from the depression subreddit and 9901 messages from the randomized control sample. Preprocessing included the removal of links, emoticons, messages tagged as 'removed' or 'deleted,' and empty text messages. In addition, any posts by users who posted more than one message or cross-posted in both conditions were removed. The final sample size consisted of 8690 messages (5703 from the depression forum condition). Passive voice and CCR loadings were extracted the same way as in Studies 1-2.

Similarly to Studies 1 and 2, a rater again coded 100 texts from this sample. The rater counted instances of non-agentive language. Spearman's rank correlation coefficient of the rater's coding and the occurrences of passive voice derived from spaCy was positive and significant, $r = 0.57$, $p < 0.001$. Once again, we see this as evidence supporting the use of spaCy to quantify non-agentive language in Reddit posts.

Study 3a was pre-registered at https://aspredicted.org/92i74.pdf, however, upon inspection of the data, it was evident that despite the lengthy nature of the message, passive voice was heavily zero-inflated, such that using a linear model on normalized counts was the wrong analytical choice. In light of this, we deviated from the pre-registered plan and analyzed the data using a count model; given this substantial deviation, we pre-registered a replication study (Study 3b). Following suggestions from a reviewer, we conducted an analysis that included self-referential language, which was not pre-registered in either study. For full transparency, we report the analyses aligning with the second pre-registration in the SI (Tables S11 and S12).

Study 3b was an exact replication of Study 3a, with the exception that the collected data were posted between November 2016 and July 2019. The original sample size consisted of 9999 messages from the depression subreddit, and 10,001 messages from the control sample. After preprocessing, the final sample

size consisted of 9685 (6325 from the depression forum condition). The replication was pre-registered as well at https://aspredicted.org/rv3iq.pdf. The study was conducted under the Departmental IRB guidelines and was ruled "exempt".

Study 3c was a replication of Studies 3a and 3b, with a different set of control subreddits. To make a comprehensive list of support groups that do not mainly provide emotional support, we prompted ChatGPT4 chatbot[64] with the following; "please generate a list of 200 actual reddit forums. make sure to adhere to the following criteria: 1. make sure the forum provides support or assistance. 2. make sure the forum actually exists. 3. make sure the forum doesn't deal with psychological content matter or emotionally-charged topics. 4. make sure that you do not repeat yourself and generate duplicate entries". This prompt generated 94 subreddits (see Supplementary Note 3 in SI).

The data for Study 3c was obtained by querying a vast dataset of pre-collected Reddit activity[63], which is readily accessible on BigQuery, a data warehouse platform developed by Google that facilitates the manipulation of data on a large scale. We collected data from the entire month of June 2019, querying the depression subreddit and the list of 94 subreddits of interest. We were able to collect data from 70 support groups and 79 control subreddits. Preprocessing was done in the same fashion as Studies 3a and 3b. Upon finalizing data collection and preprocessing, we were left with 8255 unique posts from the depression subreddit; therefore we sampled 8255 posts from the support groups, and another sample of 8255 which was derived from 79 control subreddits; $N = 24,765$. The study was conducted under the Departmental IRB guidelines and was ruled "exempt". CCR embeddings in this study were extracted in Python via the SentenceTransformers framework[59].

For all statistical models reported in the paper, we have checked that the data meet the assumptions of the model, primarily via posterior predictive checks and visualization. Note that our main modeling approach relies on a generalized linear model, which does not have formal tests for many assumptions.

**Reporting summary.** Further information on research design is available in the Nature Portfolio Reporting Summary linked to this article.

## Results

**Study 1.** We combined the data gathered from mTurk and Prolific to a unified dataset of $N = 835$. For a passive voice histogram, see Figure S1. We fitted a negative binomial count model, predicting the number of passive auxiliary verbs by power condition (high vs. low), self-referential language, and keeping text length and text source (mTurk/Prolific) as covariates (self-referential language and word count were median-centered across all studies).

We first examined whether the experimental manipulation affected our measures of personal agency. Participants in the high power condition ($N = 423$) and low power condition ($N = 412$) differed in their expressed constrained sense of control [$M_H = 0.23$, $SD_H = 0.12$; $M_L = 0.25$, $SD_L = 0.11$; $t(828.62) = -2.28$, $p = 0.023$, Cohen's $d = -0.16$, 95% CI -[0.29, −0.02]], locus of control [$M_H = 0.032$, $SD_H = 0.07$; $M_L = 0.002$, $SD_L = 0.07$; $t(831.96) = 6.18$, $p < 0.001$, Cohen's $d = 0.43$, 95% CI [0.29, 0.57]], and depression [$M_H = 0.35$, $SD_H = 0.11$; $M_L = 0.40$, $SD_L = 0.12$; $t(827.91) = -6.90$, $p < 0.001$, Cohen's $d = -0.48$, 95% CI [−0.62, −0.34]] in the expected directions, as revealed by their textual responses.

Turning to the main analysis, we found that the low power condition was associated with a 65% increase in passive voice, IRR (incidence rate ratio) = 1.65, $p < 0.001$, 95% CI [1.35, 2.02]

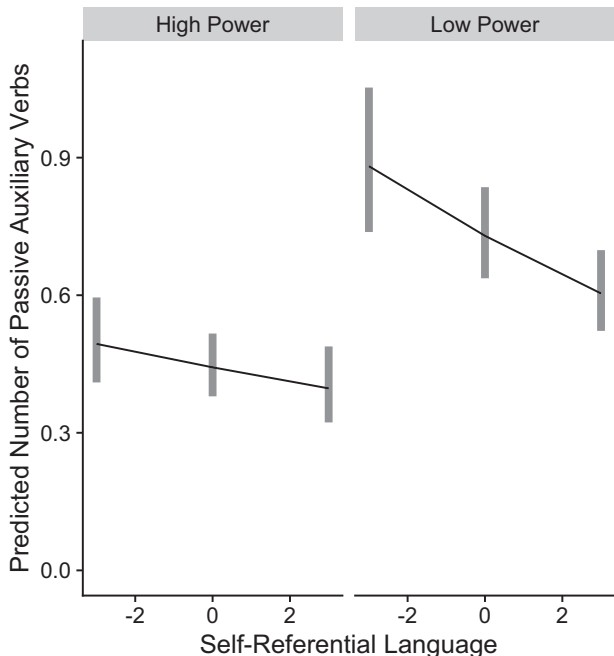

**Fig. 1 Predicted passive voice use by condition in Study 1 (*N* = 835).** *X* axis represents self-referential language (median-centered; showing interquartile range), *Y* axis represents the predicted number of passive auxiliary verbs, and panels represent the condition groups. Lines denote marginal group means; gray rectangles represent 95% confidence intervals. Low power condition is associated with 65% increase in passive voice.

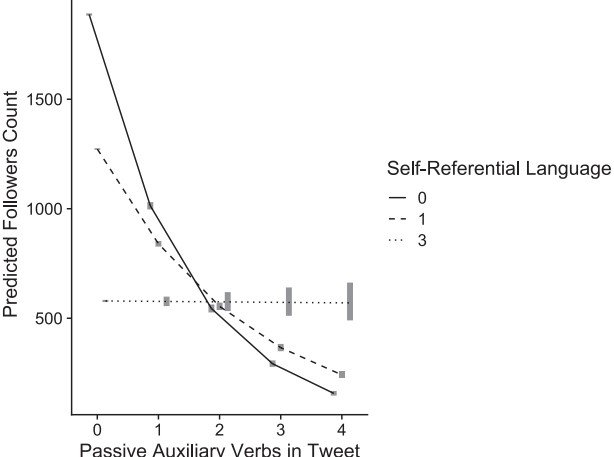

**Fig. 2 Predicted followers count by passive voice in Study 2.** Predicted followers count as a function of the number of passive auxiliary verbs in the tweet and self-referential language (0: solid line; 1: dashed line; 3: dotted line; 0–3 range covers more than 99% of the distribution; *N* = 2,726,733). Lines denote marginal means; gray rectangles represent 95% confidence intervals.

see Fig. 1 and Tables S1, S2 in Supplementary Information (SI). We did not find credible evidence for an effect of text source IRR = 0.94, *p* = 0.498, 95% CI [0.78, 1.13], nor did we find credible evidence for an effect of self-referential language IRR = 0.96, *p* = 0.082, 95% CI [0.92, 1.01].

When considering self-referential language as the dependent variable, we found that the low power condition was associated with a 29% increase in self-referential language, IRR = 1.29, *p* < 0.001, 95% CI [1.22, 1.36], supporting the reduced self-agency hypothesis (see table S3 in SI).

The results of Study 1 show greater use of non-agentive language when participants describe incidents wherein other people had control over them, vs. incidents where they were the ones with control over others. These findings provide initial evidence for the link between personal and linguistic agency, and suggest that reductions in sense of personal agency are reflected in reductions in linguistic agency. The association between passive voice and self-referential language was negative in its direction, however, it did not reach statistical significance.

Taken together, manipulating power in a controlled environment leads to changes in linguistic markers of agency; however, a question remains whether such a relationship occurs naturally in ecological settings. To better understand these links, Study 2 provides an analysis of linguistic agency and infulence on social media.

**Study 2**. To examine the relation between the number of followers and passive voice on social media, we first aggregated our twitter sample by users, to avoid dependencies in the model. For each user in the sample, we calculated the average use of passive voice and average Twitter followers (a user may gain followers over the course of the sampling duration); the number of followers was rounded to the nearest integer. Our final dataset consisted of 2,726,733 unique users.

We fitted a negative binomial generalized linear model predicting Twitter followers by their corresponding average passive voice use, average self-referential language, and average tweet length (as a control variable). We found that for every addition in passive auxiliary verbs the model predicts a decrease of 46% in followers count (please note that such an association does not mean that this is a causal effect), IRR = 0.54, *p* < 0.001, 95% CI [0.53, 0.55], self-referential language was negatively associated with followership, IRR = 0.67, *p* < 0.001, 95% CI [0.67, 0.68], and that the link between passive auxiliary verbs and followership was moderated by the degree of self-referential language, IRR = 1.23, *p* < 0.001, 95% CI [1.21, 1.25]. See Fig. 2 and Table S4 in SI.

To support the passive voice analysis, we extracted CCR embeddings from a sample of 100,000 tweets. After addressing dependencies, our final sample comprised 81,606 unique users. Although this sample is relatively large, it accounts for <1% of the original sample size. Because it is more sensitive to outliers, and because it is computationally feasible to do so, we opted to perform a robust generalized linear regression. To alleviate the computational load, we employed bootstrapping, running a robust negative binomial model 1000 times, with each iteration modeling 1000 unique users at a time. Our results indicate that all coefficients are in the hypothesized direction, however, 95% confidence intervals do not exclude IRR = 1. When considering constrained sense of control: average IRR = 0.33, 95% CI [0.04, 1.21]; internal vs. external locus of control: average IRR = 3.61, 95% CI [0.18, 18.89]; depression: average IRR = 0.31, 95% CI [0.04, 1.22]. A more lenient confidence band shows the methods do converge in constrained sense of control (90% CI [0.06, 0.90]) and depression (90% CI [0.05, 0.86]) but not in locus of control (90% CI [0.25, 13.90]).

Study 2 joins the results of Study 1 in showing that linguistic and personal agency are correlated in natural language use. Across 2.7 million users, greater passive voice use was associated with fewer followers (i.e., lower social rank on social media) such that each instance of passive voice is related to a decrease of 46% in followership. This finding showcases how social standing is related to agentive language use.

**Study 3**. In the current study, We leveraged data from online discussion forums that provide a space for people living

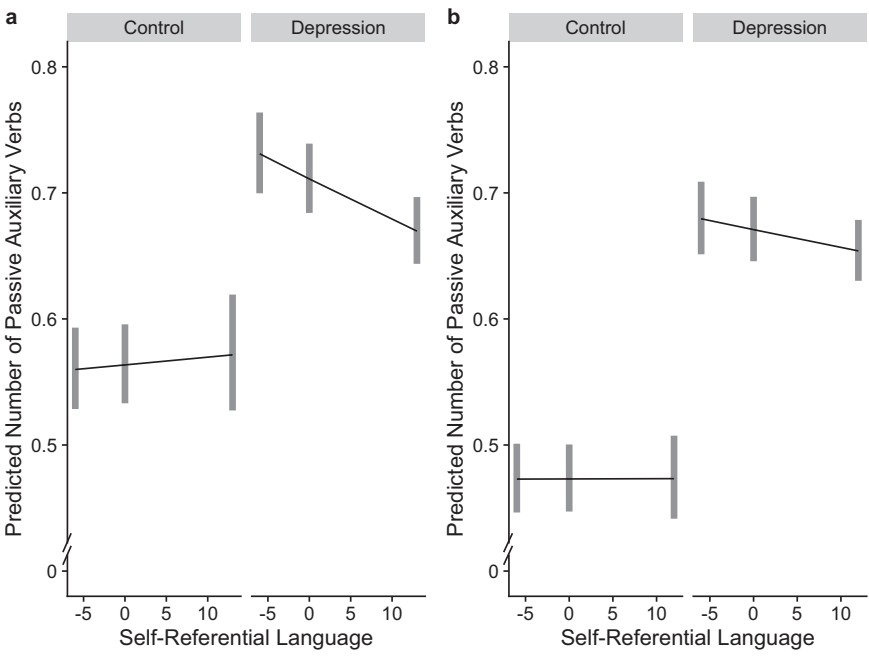

**Fig. 3 Predicted passive voice use by subreddit groups in Studies 3a and 3b.** Predicted number of passive auxiliary verbs by subreddit group and self-referential language in Study 3a (**a**; $N = 8690$) and Study 3b (**b**; $N = 9685$). X axis shows median-centered, interquartile range of self-referential language. Lines denote marginal means; gray rectangles represent 95% confidence intervals.

with depression to investigate the hypothesis that people who experience depression increase their use of non-agentive language.

We fitted a negative binomial generalized linear model predicting the number of passive auxiliary verbs by group (depression forum vs. control forums) and keeping word count as a covariate. We found that the language in the depression forum (vs. control forums) was associated with a 26% increase in passive voice use, $IRR = 1.26$, $p < 0.001$, 95% CI [1.18, 1.35]. There was no credible evidence that self-referential language affected the use of passive voice, $IRR = 1.00$, $p = 0.592$, 95% CI [0.99, 1.01], but there was a significant interaction $IRR = 0.99$, $p < 0.001$, 95% CI [0.98, 0.99]. See Fig. 3 and Table S5 in SI.

When self-referential language was modeled as the variable of interest, we found that the depression forum group had more than double self-referential expressions vs. the control group, $IRR = 2.54$, $p < 0.001$, 95% CI [2.45, 2.62], see Table S6 in SI.

Similarly, CCR analyses showed the expected pattern of results. Users in the depression forum and control forums differed in their expressed constrained sense of control [$M_{Dep} = 0.45$, $SD_{Dep} = 0.14$; $M_{Control} = 0.10$, $SD_{Control} = 0.11$; $t(7230.5) = 129.35$, $p < 0.001$, Cohen's $d = 2.74$, 95% CI [2.68, 2.80]], locus of control [$M_{Dep} = -0.04$, $SD_{Dep} = 0.07$; $M_{Control} = 0.01$, $SD_{Control} = 0.08$; $t(5922.8) = -29.10$, $p < 0.001$, Cohen's $d = -0.66$, 95% CI [-0.71, -0.62]], and depression [$M_{Dep} = 0.46$, $SD_{Dep} = 0.13$; $M_{Control} = 0.15$, $SD_{Control} = 0.12$; $t(6152.8) = 110.38$, $p < 0.001$, Cohen's $d = 2.48$, 95% CI [2.42, 2.54]] in the expected directions, as revealed by posts on Reddit.

In line with our pre-registered prediction, these results show that people who participate in a depression forum use passive voice to a greater extent. Given the deviation from the pre-registered plan, we ran a pre-registered replication of Study 3a in which we collected older data, one year prior.

In line with our pre-registered hypothesis and the results of Study 3a, we found that the language in the depression forum was associated with a 42% increase in passive voice use, $IRR = 1.42$, $p < 0.001$, 95% CI [1.32, 1.52]. There was no main effect for self-referential language, $IRR = 1.00$, $p = 0.978$, 95% CI [1.00, 1.00],

and no significant interaction, $IRR = 1.00$, $p = 0.094$, 95% CI [0.99, 1.00]. See Fig. 3 and Table S7 in SI.

When considering self-referential language, we found a strikingly similar pattern of results: the depression forum group had significantly more self-referential expressions vs. the control group, $IRR = 2.56$, $p < 0.001$, 95% CI [2.48, 2.65], see Table S8 in SI.

Once again, CCR analyses provide results in the expected direction. Users in the depression forum and control forums differed in their expressed constrained sense of control [$M_{Dep} = 0.45$, $SD_{Dep} = 0.14$; $M_{Control} = 0.09$, $SD_{Control} = 0.10$; $t(8480.6) = 140.42$, $p < 0.001$, Cohen's $d = 2.77$, 95% CI [2.52, 2.83]], locus of control [$M_{Dep} = -0.04$, $SD_{Dep} = 0.07$; $M_{Control} = 0.02$, $SD_{Control} = 0.07$; $t(6897.5) = -34.08$, $p < 0.001$, Cohen's $d = -0.73$, 95% CI [-0.77, -0.68]], and depression [$M_{Dep} = 0.46$, $SD_{Dep} = 0.12$; $M_{Control} = 0.13$, $SD_{Control} = 0.12$; $t(7059) = 125.23$, $p < 0.001$, Cohen's $d = 2.64$, 95% CI [2.59, 2.70]] in the expected directions, as revealed by posts on Reddit.

In Study 3b we found and replicated that non-agentive language was much more prevalent (up to 42% more) in a depression-related community vs. a random sample of Reddit communities. These findings suggest that real-life situations that involve a diminished sense of control and agency are strongly related to diminished linguistic agency. However, some may argue that our choice of control subreddits was not optimal because the depression subreddit predominantly prioritizes providing peer support, which may have its own unique linguistic structure. Therefore, we ran another replication (Study 3c) where we constructed a set of subreddits that are devoted to supporting and assisting others in topics unrelated to psychological help (e.g., cooking advice, programming support, etc.).

In line with the results of Studies 3a and 3b, we found that the language in the depression forum was associated with a 16% increase in passive voice use, $IRR = 1.16$, $p < 0.001$, 95% CI [1.11, 1.22], whereas the support groups had 10% increase of passive voice, $IRR = 1.10$, $p < 0.001$, 95% CI [1.06, 1.16]. There was a main effect for self-referential language, such that it was negatively associated with passive voice $IRR = 0.99$, $p < 0.001$,

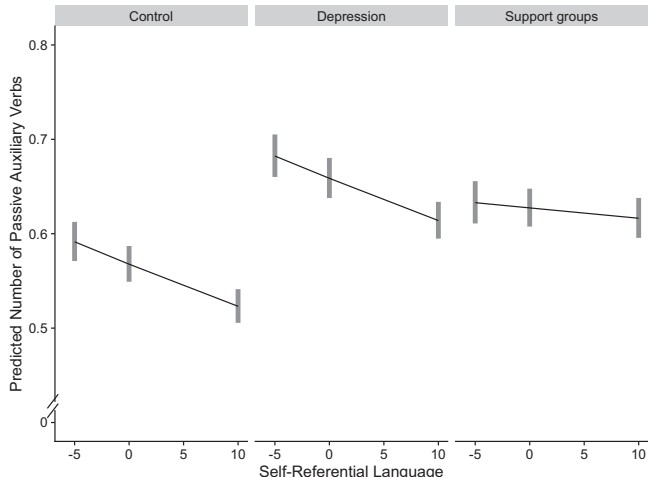

**Fig. 4 Predicted passive voice use by subreddit groups in Study 3c.**
Predicted number of passive auxiliary verbs by subreddit group and self-referential language in Study 3c ($N = 24{,}765$). X axis shows median-centered, interquartile range of self-referential language. Lines denote marginal means; gray rectangles represent 95% confidence intervals.

95% CI [0.99, 0.99], and a significant interaction. See Fig. 4 and Table S9 in SI for full interaction details.

A planned comparison between the depression forum and support groups revealed that passive voice use was significantly greater in the depression forum group vs. support groups, IRR = 1.05, $p < 0.001$, 95% CI [1.01, 1.10].

When self-referential language was considered, the pattern of results remained consistent: posts in depression forum were two times more likely to involve self-referential language, IRR = 2.06, $p < 0.001$, 95% CI [2.01, 2.11], whereas the support groups were associated with a 7% increase in self-referential language, IRR = 1.07, $p < 0.001$, 95% CI [1.04, 1.09], (see Table S10 in SI). A planned comparison between the depression forum and support groups revealed that self-referential use was significantly greater in the depression forum group vs. support groups, IRR = 1.93, $p < 0.001$, 95% CI [1.88, 1.98].

CCR analyses confirmed the expected results, with the depression forum group differing significantly from the control and support group conditions [constrained sense of control: $F(2, 24{,}762) = 18148.17$, $p < 0.001$, $\eta^2 G = 0.594$, 95% CI [0.59, 0.59]; locus of control: $F(2, 24{,}762) = 776.62$, $p < 0.001$, $\eta^2 G = 0.059$, 95% CI [0.05, 0.06]; depression: $F(2, 24{,}762) = 14122.16$, $p < 0.001$, $\eta^2 G = 0.533$, 95% CI [0.53, 0.53]]. See Fig. S2 in SI.

## Discussion

In the current work, we examined the relation between linguistic and personal agency. In Study 1 we found that a manipulation of participants' sense of personal agency (specifically, of sense of power) affected their use of agentive language. In Studies 2 and 3 we used text analyses in ecological contexts to show that (i) increased personal agency (operationalized as one's social media "influence") is associated with more agentive language; (ii) that the language used in a forum for individuals concerned with depression and therefore may experience a diminished sense of agency is less agentive.

It is often argued that the use of agentive language is related to a sense of individual agency and control[8,10]. However, past research has not provided direct quantitative evidence for this claim. Previous work has mostly focused on examining how linguistic agency influences our attribution of personal agency to others, and specifically, how it affects attributions of blame[19,65,66]. When it comes to the relation between individuals' own sense of personal agency and linguistic agency, previous studies have

mostly utilized qualitative analyses[27]. In the current work, we present a comprehensive study of the relation between linguistic and personal agency, corroborating the longstanding suggestions of their supposed interrelatedness.

Study 1 involved an experimental manipulation, allowing us to substantiate the causal effect of sense of personal agency on linguistic agency. In Studies 2 and 3, we examined the relation between linguistic and psychological agency in naturalistic settings. Specifically, we saw passive voice is negatively associated with gaining followers on Twitter, such that for every passive auxiliary verb the model predicted a 46% reduction in followership, directly linking linguistic agency with social rank and control. Furthermore, we observed that online communities of people experiencing depression use up to 42% more passive voice than a random sample of online communities, suggesting that lower levels of linguistic agency may be a marker of deteriorated mental health. Notably, these results show that subtle variation in linguistic agency is related to matters of clinical and social importance.

While our main analysis pertained to the use of the passive voice, we also conducted an exploratory investigation of how personal agency is related to people's use of self-referential language. On the one hand, the perception of self-agency entails that the self is the causer of events; as such, one might expect that I-language will be correlated with higher personal agency. On the other hand, self-reference is a known correlate of depression, which is characterized by a markedly reduced sense of self-agency, and as such may be associated with lower personal agency. Across all three studies, lower personal agency (i.e., lack of control, lower social rank, and depression forum participation) was associated with greater use of self-referential language. These findings replicate the well-known relation between personal pronoun use and depression[53], and highlight that this relation generalizes to broader lower agency states (e.g., lower social rank, lack of control).

**Limitations**. One issue that needs to be addressed in future studies is whether the associations observed in Studies 2 and 3 reflect a stable (trait-level) or state-level phenomenon. For example, a person may feel chronically disempowered in their daily lives but may feel empowered in the virtual world—whenever they address a large group of interested followers. Likewise, a person who experiences a depressive episode may lose their sense of agency, but regain it once their mood stabilizes. In addition, while Study 1 benefits from an experimental design, Studies 2 and 3 are correlational and limit our causal inference on whether people who use more active language accrue a greater following or whether an increased sense of control leads to more agentive language use. Importantly, Study 3 draws a connection between participation in a depression-related online forum and the expression of depressive experiences. It is essential to underscore that there exists no available data regarding the clinical or sub-clinical depression scores of the individuals who authored the threads within the depression subreddit. The attribution of depression status is solely deduced from their engagement within the forum. Therefore, this study should be considered as yielding more circumstantial evidence than the others. Further research using non-correlational designs will be required to address these questions.

It is important to note that while our findings suggest a relationship between personal agency and linguistic agency, the reported effects may not be exclusively driven by personal agency. The predictors examined in our study, such as sense of power, number of followers, and depression, share the common characteristic of personal agency[39,40,67–69]. However, other factors and aspects related to these predictors may also contribute to the observed effects on linguistic agency. For instance,

depression encompasses a wide range of symptoms and experiences beyond a lack of agency, and the number of followers may reflect various aspects of an individual's online presence, such as social status or content quality. Similarly, the sense of power can be influenced by numerous personal and situational factors that may affect linguistic agency independently. Thus, while personal agency is a shared characteristic of all predictors, we acknowledge the possibility that the reported effects are not all exclusively driven by personal agency. In other words, when considering each study individually, each one focuses on specific aspects of the complex connection between language and personal agency. However, when we examine these studies collectively, we believe that they contribute to our understanding of how language reflects individuals' sense of personal agency.

An additional limitation of the current research is that it examines the relation between linguistic and personal agency only in a single cultural-linguistic context. While our current study does not provide direct evidence for this, we hypothesize that the relationship between personal agency and linguistic agency could potentially be observed in other languages as well, considering the universal nature of the concept of agency. However, the specific linguistic features that signal agency might differ across languages due to grammatical and cultural variations[70].

## Conclusions

The current research sets the stage for future research that examines culture-level variations in linguistic and psychological agency. The degree to which cultures differ in their prevailing beliefs about one's sense of control has important societal consequences including economic development[71–74] and upward mobility[75]. Future research could adopt a cross-cultural perspective to examine whether cultures whose language prefers agentivity are more likely to adopt agentic beliefs[76,77], thereby further elucidating the consequences of (supposedly arbitrary) linguistic features on human life.

Our findings have demonstrated the interrelatedness between linguistic agency and personal agency. Specifically, we saw that variations in personal agency–due to sense of social power, social influence, and depression-related forum participation–are associated with changes in linguistic agency. By applying diverse computational text analysis methodologies, encompassing both experimental and ecological contexts from a sizable population of users and participants, the results provide us a comprehensive view of how language and thought are intertwined, in an important psychological context. This approach could be useful for future work examining the relation between language and thought, and may form the basis for studies that examine how linguistic interventions change people's sense of agency.

## Data availability

All shareable data are found on the online OSF repository at https://osf.io/nwsx3/.

## Code availability

All the code including reproducible analyses are found on the online OSF repository at https://osf.io/nwsx3/.

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

## Acknowledgements

We thank Roey Gafter, Yoav Kessler, Yoav Bar Anan, Ruvi Dar, Si Berrebi, Noa Bassel, Or Kronenblum, Almog Bowman, Ofir Neomi Aharon, Chaya Leibman, Yuval Gaiger, Coral Rosental, and Bore' Olam. This research was supported by the United States - Israel Binational Science Foundation grant no. 2015258 to M.G; Israel Science Foundation grant no. 1113/18 to M.G. and by the Ministry of Science & Technology, Israel. The funders had no role in study design, data collection and analysis, decision to publish, or preparation of the manuscript.

## Author contributions

Conceptualization by A.S. and M.G.; Data curation by A.S and M.G.; Formal analysis by A.S.; Funding acquisition by M.G. and A.S.; Investigation by A.S., B.H., and M.G.; Methodology by A.S. and M.G; Software by A.S.; Supervision by M.G.; Validation by A.S..; Visualization by A.S.; Writing by A.S., B.H., and M.G.

## Competing interests

The authors declare no competing interests.
