## [Peer Review File · Communications Psychology]

Web links to the author's journal account have been redacted from the decision letters as indicated to maintain confidentiality.

13th Apr 23

Dear Dr Simchon,

Thank you for your patience during the peer-review process. Your manuscript titled "On Pulling the Strings or Having Your Strings Pulled: The Relation Between Personal and Linguistic Agency" has now been seen by 3 reviewers, and I include their comments at the end of this message. They find your work of interest, but raised some important points. We are interested in the possibility of publishing your study in Communications Psychology, but ask for some key concerns to be addressed before we can consider your manuscript further.

We therefore invite you to revise and resubmit your manuscript, along with a point-by-point response to the reviewers. Please highlight all changes in the manuscript text file.

Editorially, we ask you to address the reviewers' requests for additional empirical evidence to strengthen your case (all reviewers mention this, but especially Reviewer #2).

We also ask you to revise the presentation of your findings. As reviewer #3 highlights, there are questions as to whether all three studies measure the same concept. Please add nuance to the presentation of how the the measure from each study informs the overarching concept. Please appropriately caveat how much each study may consequently inform the central research question. Please also remove all claims of causality (e.g., lines 212-214).

Similarly following the referees' advice, we ask you to discuss the relevant literature in more depth. The reviewers give some valuable suggestions that may help you to strengthen your work by putting it in context with previous research, although there is no requirement to include each individual reference.

In addition, the revised manuscript needs to contain the following information, even where it is already contained in the reporting summary:

- Information on whether ethics had been obtained for each study
- Information on whether informed consent had been obtained for Studies 3 a & b
- Information on whether the Twitter data was obtained through Twitter's dedicated API
- Information on how the Reddit data was obtained and the terms of use on which the data acquisition rests;
- Disclosure of all deviations from the pre-registered protocol, for more information go here: <https://www.nature.com/commpsychol/submit/preregistration>
- Data and code availability statements, see here: <https://www.nature.com/commpsychol/submit/submission-guidelines#data-availability>
- Revised title: please choose a descriptive title that captures your main results

Please use the following link to submit your revised manuscript, point-by-point response to the referees' comments (which should be in a separate document to any cover letter) and the completed checklist:

[link redacted]

We hope to receive your revised paper within 8 weeks; please let us know if you aren't able to submit it within this time so that we can discuss how best to proceed. If we don't hear from you, and the revision process takes significantly longer, we may close your file. In this event, we will still be happy to reconsider your paper at a later date, provided it still presents a significant contribution to the

literature at that stage.

Please do not hesitate to contact me if you have any questions or would like to discuss these revisions further. We look forward to seeing the revised manuscript and thank you for the opportunity to review your work.

Best regards,

Antonia Eisenkoeck

Antonia Eisenkoeck
Senior Editor
Communications Psychology

EDITORIAL POLICIES AND FORMATTING

Editorial Policy: [Policy requirements](https://www.nature.com/documents/nr-editorial-policy-checklist.pdf) (Download the link to your computer as a PDF.)

Furthermore, please align your manuscript with our format requirements, which are summarized on the following checklist:

[Communications Psychology formatting checklist](https://www.nature.com/documents/commsj-style-formatting-checklist-review-perspective.pdf)

and also in our style and formatting guide [Communications Psychology formatting guide](https://www.nature.com/documents/commspsychol-style-formatting-guide-accept.pdf) .

* **CODE AVAILABILITY:** All Communications Psychology manuscripts must include a section titled "Code Availability" at the end of the methods section. In the event of publication, we require that the custom analysis code supporting your conclusions is made available in a publicly accessible repository; at publication, we ask you to choose a repository that provides a DOI for the code; the link to the repository and the DOI will need to be included in the Code Availability statement. Publication as Supplementary Information will not suffice. We ask you to prepare code at this stage, to avoid delays later on in the process.

* **DATA AVAILABILITY:**

All Communications Psychology manuscripts must include a section titled "Data Availability" at the end of the Methods section or main text (if no Methods). More information on this policy, is available at <http://www.nature.com/authors/policies/data/data-availability-statements-data-citations.pdf>.

At a minimum the Data availability statement must explain how the data can be obtained and whether there are any restrictions on data sharing. Communications Psychology strongly endorses open sharing of data. If you do make your data openly available, please include in the statement:

We recommend submitting the data to discipline-specific, community-recognized repositories, where possible and a list of recommended repositories is provided at <http://www.nature.com/sdata/policies/repositories>.

If a community resource is unavailable, data can be submitted to generalist repositories such as [figshare](https://figshare.com/) or [Dryad Digital Repository](http://datadryad.org/). Please provide a unique identifier for the data (for example a DOI or a permanent URL) in the data availability statement, if possible. If the repository does not provide identifiers, we encourage authors to supply the search terms that will return the data. For data that have been obtained from publicly available sources, please provide a URL and the specific data product name in the data availability statement. Data with a DOI should be further cited in the methods reference section.

REVIEWERS' EXPERTISE:

Reviewer #1: natural language processing tools & social media text data
Reviewer #2: psycholinguistics, social power, natural language processing
Reviewer #3: personal agency, social power

REVIEWERS' COMMENTS:

Reviewer #1 (Remarks to the Author):

This interesting paper connects the use of non-agentive language to social and personal agency. This is a well-written work describing three well-designed studies. I really like the paper and look forward to seeing it in print. Below, are a few suggestions that the author can consider in revising their work.

1. In all studies, the authors use a top-down measure of agentive language. Theoretically, this makes a lot of sense. But there could be interesting bottom-up or data-driven patterns as well, which could potentially shed light on previously unknown linguistic signatures of personal agency. The authors can conduct a simple topic modeling (e.g., LDA; see Maier et al., 2018) to examine what topics (or "themes") high-agency (vs. low-agency) people talk about in Studies 2 and 3 (on Twitter and Reddit).
2. In the top-down analysis of agentive language, the authors only use one kind of analysis. It would be interesting to conduct additional robustness checks using different top-down methods. For example, the authors can use the Contextualized Construct Representation method (CCR; Atari et al.,

2023) – relying on a validated psychometric measure of agency, power, locus of control – to quantify agentive language. It is generally a good idea to use multiple methods when examining the relationship between language use and psychological ground truth (for a discussion, see Kennedy et al., 2021).

3. I think one important limitation of the current work is that it is based on one culture (i.e., American) and one language (i.e., English; see Blasi et al., 2023). This is fine, and the authors do touch on this toward the end of their Discussion, but in my opinion, it should be more explicitly voiced in the paper. Can it be the case that these findings are WEIRD (Henrich et al., 2010)? Do the authors expect to see the same pattern in non-English languages?

Minor points:

I am really glad that the authors have provided annotations and some benchmarking. The Validation sections are rather short, though. It would be great if the authors could add more details to the Validation sections.

Please rephrase terms like “using big data”, “big data analyses”, and “big-data methods” as the term “big data” is vague with no added value. Instead, describe the exact nature of the method used (e.g., using terms like language analysis, text analysis, NLP, etc.).

References

Atari, M., Omrani, A., & Dehghani, M. (2023, February 24). Contextualized Construct Representation: Leveraging Psychometric Scales to Advance Theory-Driven Text Analysis. <https://doi.org/10.31234/osf.io/m93pd>

Blasi, D. E., Henrich, J., Adamou, E., Kemmerer, D., & Majid, A. (2023). Over-reliance on English hinders cognitive science. *Trends in cognitive sciences*.

Henrich, J., Heine, S. J., & Norenzayan, A. (2010). The weirdest people in the world?. *Behavioral and brain sciences*, 33(2-3), 61-83.

Kennedy, B., Atari, M., Davani, A. M., Hoover, J., Omrani, A., Graham, J., & Dehghani, M. (2021). Moral concerns are differentially observable in language. *Cognition*, 212, 104696.

Maier, D., Waldherr, A., Miltner, P., Wiedemann, G., Niekler, A., Keinert, A., ... & Adam, S. (2018). Applying LDA topic modeling in communication research: Toward a valid and reliable methodology. *Communication Methods and Measures*, 12(2-3), 93-118.

Signed review,
Mohammad Atari

Reviewer #2 (Remarks to the Author):

This review is for the manuscript "On Pulling the Strings or Having Your Strings Pulled: The Relation Between Personal and Linguistic Agency." Overall, there are many things to like about this manuscript: it is well-written, direct, and it is clear that the research was conducted thoughtfully. As I was reading this manuscript, two primary thoughts came to mind that I feel would help to strengthen it in terms of its appeal to a broader audience, as well as help to disambiguate some of the constructs that may be in play.

Major points of feedback:

1. One of the big things that I think would strengthen the manuscript would be to connect with a broader set of literatures that, historically, have been quite interested in understanding the psychology that underpins the use of active/passive voice, as well as other studies that show the consequences of framing thought in this fashion. The authors note in the manuscript that a majority of research on active/passive voice and agency deals with the constructs of person perception/attribution. I absolutely agree with this sentiment, although, I might suggest that this is a slightly narrow view that is at the intersection of the fields of social psychology and psycholinguistics. Historically — at least through much of the 20th century if not before — psychotherapists, clinical psychologists, and discourse analysts have focused considerable attention on the relationship between linguistic agency and personal agency. Much of the work that I'm familiar with in these spaces is, I think, rooted largely in assumptions about the correspondence between the two, often deriving evidence for their relationship from clinical case studies and smaller samples. While none of this detracts from the current work in and of itself, I might suggest that the authors broaden the scope of the introduction/discussion sections leverage the bodies of scholarship, practice, and academic thought in this area. I'll provide some references at the end of this review that may (or may not!) serve as helpful guideposts to some of the literature that I'm describing here. Note that I'm not suggesting that the authors need to cite all, or even any, of the references that I provide — rather, I might instead recommend them as examples that highlight a tradition of acknowledging the importance of the common ground between linguistic agency and personal agency, often in terms of understanding pretty central topics such as personal behavior, psychosocial change, mental health, emotion, and relationships. I'll also note that I'm passingly aware of a literature in criminology/justice/law that dovetails with some of the ideas raised in the existing literature, however, I'll admit that I'm far less familiar with work in these areas and, additionally, their relevance to the current research may not be as strong.

To wrap up my thought here: given the importance that is often placed on active-vs-passive voice in therapeutic contexts, connecting with this and other broader/richer traditions might expand the scope of the current work and, additionally, facilitate greater impact within several major areas of research and real-world outcomes.

2. Related to the above, one big set of concepts that might be meaningful to delve into and disambiguate with the current set of studies are those of psychological distance, immediacy, and internal-vs-external focus. I'm treating all of these as fairly unified, conceptually speaking (although I know that there are nuances in how they are studied, understood and described), in part, because I believe that the heart of a lot of these and related constructs generally relate to a focus on oneself versus the environment (or, sometimes, the relationship/distance between the two). There is a decent amount of work that suggests that active-vs-passive framing is primarily (or at least, in part) a reflection of whether a person is focusing on oneself or something in the environment (e.g., another person, an object in the environment, etc.). For example, some work shows that active-vs-passive framing is related to goal attainment (e.g., focusing on one's own hunger vs. the qualities of the food being consumed in a weight loss context; "I ate the cake" versus "the cake was eaten by me"). It might be possible, then, to characterize or consider at least some of the current findings through a lense of "self versus external." For example, Study 1's results may not only reflect a sense of personal agency, but might also be a function of prompting a person's focus on their power versus the power of other individuals.

A potentially interesting way to help disambiguate personal agency from self/other would be to do an additional check on the use of first-person singular pronouns as subject (I, I'm, I'll, etc.) versus object (e.g., "me"). Do high and low linguistic agency correspond to self-as-subject and self-as-object, respectively? I don't have a sense of how this might work in the context of the Twitter data, as many of the tweets may not involve the self at all (e.g., "This website is having a sale on socks!"), but my

intuition is that the self-narratives in Study 1 and the r/depression subreddit posts would be likely to invoke the self quite a bit. Or, alternatively (and perhaps more simply) — is there an inverse relationship between passive voice and self-referential language? My sense is that, regardless of the results, this might be an interesting thing to report that would also help to contextualize the current findings and lend to a more rich and informative interpretation of what is happening, psychologically speaking, when a person uses the passive voice.

Smaller thoughts:

3. One thing that I was curious about was the nature of the data for both Twitter and Reddit: is it the case that all of the texts were "top level" posts? For example, were all of the Reddit posts analyzed "submissions" rather than comments in response to submissions? Likewise, did the tweet corpus include both initial tweets as well as responses/comments to other tweets? Going back to the idea of self-versus-other focus, I can imagine a bit of difficulty in separating the idea of "I'm tweeting in response to another person" and the use of passive language, which might reflect "I'm focusing on you" (whereas a top-level tweet might be more likely to be about oneself, and use less passive voice). It might also be the case that people who have more followers would be more likely to make, rather than respond to, other tweets.

I don't think that any of those ideas need to be fleshed out in the paper itself, but it might be helpful to provide a bit of additional information (in supplementary materials, perhaps?) about the data itself. And, of course, if this information is already present and I missed it, please accept my apologies!

4. Again, a very minor thought: I would be curious to know whether submissions to the various subreddits receive more comments or upvotes if they use less passive language, which might help reinforce the findings from Study 2. Again, nothing that I would necessarily include in the manuscript itself unless the authors feel that it strengthens the paper, and this is definitely not a major thing that I think even needs to be explored if the authors feel that it's too tangential.

5. For the depression subreddit findings, one concern that I have is that the depression subreddit is fairly unique from most (perhaps all?) of the comparison subreddits in that it is generally framed as a place for peer support & help-seeking. While the assumption is fair that the experience of depression is the major contributor to the findings here, I might suggest a more analogous set of comparison groups than a random selection of subreddits. I do know that it can be difficult to find support-seeking groups where there is a low likelihood of comorbidity with depression (for example, people suffering from medical conditions are likely also going to be under emotional duress), there may be several subreddits that focus on support/help-seeking that are less affectively tinged (e.g., r/cartalk is a subreddit for help with car trouble, r/personalfinance for people looking for financial help and advice, r/fashionadvice for people looking for fashion feedback, etc.).

6. Lastly, and another very minor note: the current title of "On Pulling the Strings or Having Your Strings Pulled" rang a bit empty for me in terms of signalling the studies in the manuscript. Initially, it read to me more along the lines of classic findings in the way of person perception/attribution studies — is a person responsible for their actions, or was it some external force? Of course, the phrasing is relevant to the active/passive distinction, but I might suggest something that more clearly relates to whether a person is high/low in personal agency rather than something that might be interpreted as an attributional cause of one's actions.

In sum, I want to re-emphasize that I found this to be a very nice and interesting manuscript, and I hope that the authors find this feedback helpful!

Sincerely,
Ryan L. Boyd

Additional references:

Vroman, K., Warner, R., & Chamberlain, K. (2009). Now let me tell you in my own words: Narratives of acute and chronic low back pain. *Disability and Rehabilitation*, 31(12), 976–987. <https://doi.org/10.1080/09638280802378017>

Adler, J. M., & McAdams, D. P. (2007). The narrative reconstruction of psychotherapy. *Narrative Inquiry*, 17(2), 179–202. <https://doi.org/10.1075/ni.17.2.03adl>

Quayle, E., & Taylor, M. (2002). Child pornography and the internet: Perpetuating a cycle of abuse. *Deviant Behavior*, 23(4), 331–361. <https://doi.org/10.1080/01639620290086413>

Szalai K., & László J. (2007). Az aktív és passzív igék gyakorisága a csoportjelenségek tükrében: Történelemkönyvek szövegeinek narratív pszichológiai vizsgálata NooJ tartalomelemző programmal [Measuring the frequency of active and passive verbs in historical narratives: Narrative psychological analysis of the texts of history books with the NooJ program]. *Magyar Számítógépes Nyelvészeti Konferencia*, 5, 242–249. <http://acta.bibl.u-szeged.hu/58682/>

László, J., Csertő, I., Fülöp, É., Ferenczhalmy, R., Hargitai, R., Lendvai, P., Péley, B., Pólya, T., Szalai, K., Vincze, O., & Ehmann, B. (2013). Narrative language as an expression of individual and group identity: The narrative categorical content analysis. *SAGE Open*, 3(2), 2158244013492084. <https://doi.org/10.1177/2158244013492084>

Senay, I., Usak, M., & Prokop, P. (2015). Talking about behaviors in the passive voice increases task performance. *Applied Cognitive Psychology*, 29(2), 262–270. <https://doi.org/10.1002/acp.3104>

Demetci, P., Nichols, C., Zastavker, Y. V., Stolk, J. D., Dillon, A., & Gross, M. D. (2016). Internalization and externalization in the classroom: How do they emerge and why is it important? 2016 IEEE Frontiers in Education Conference (FIE), 1–5. <https://doi.org/10.1109/FIE.2016.7757463>

Schwartzman, R., & Boger, K. E. (2019). How international students using communication centers navigate locus of control. In A. G. Mag (Ed.), *Perspectives of Arts and Social Studies: Vol. Vol. 3* (pp. 36–48). Book Publisher International. <https://doi.org/10.9734/bpi/pass/v3>

Hodge, C., Pederson, J. A., & Walker, M. (2015). How do you “like” my style? Examining how communication style influences Facebook behaviors. *International Journal of Sport Communication*, 8(3), 276–292. <https://doi.org/10.1123/IJSC.2015-0052>

Kameny, R. R., & Bearison, D. J. (2002). Cancer narratives of adolescents and young adults: A quantitative and qualitative analysis. *Children’s Health Care*, 31(2), 143–173. https://doi.org/10.1207/S15326888CHC3102_5

Ferreira, F. (1994). Choice of passive voice is affected by verb type and animacy. *Journal of Memory and Language*, 33(6), 715–736. <https://doi.org/10.1006/jmla.1994.1034>

Van Staden, C. W., & Fulford, K. W. M. (2004). Changes in semantic uses of first person pronouns as possible linguistic markers of recovery in psychotherapy. *The Australian and New Zealand Journal of*

Psychiatry, 38(4), 226–232. <https://doi.org/10.1080/j.1440-1614.2004.01339.x>

Manthei, R. J., & Matthews, D. A. (1982). Helping the reluctant client to engage in counselling. *British Journal of Guidance & Counselling*, 10(1), 44–50. <https://doi.org/10.1080/03069888208258038>

Gupta, A. (2018). The effect of mindfulness, cognitive behavior therapy and academic training on conduct and academic problems of disadvantaged children. *Journal of Psychology & Psychotherapy*, 8(4), 1–12. <https://doi.org/10.4172/2161-0487.1000347>

Hughes, C. F., Uhlmann, C., & Pennebaker, J. W. (1994). The body's response to processing emotional trauma: Linking verbal text with autonomic activity. *Journal of Personality*, 62(4), 565–585. <https://doi.org/10.1111/j.1467-6494.1994.tb00309.x>

Brady, J. P. (2018). Language in schizophrenia. *American Journal of Psychotherapy*. <https://doi.org/10.1176/appi.psychotherapy.1958.12.3.473>

Lorenz, M. (1953). Language as expressive behavior. *A.M.A. Archives of Neurology & Psychiatry*, 70(3), 277–285. <https://doi.org/10.1001/archneurpsyc.1953.02320330002001>

Lorenz, M. (1955). Expressive behavior and language patterns. *Psychiatry*, 18(4), 353–366. <https://doi.org/10.1080/00332747.1955.11023020>

Reviewer #3 (Remarks to the Author):

The present paper investigates whether there is a link between linguistic agency and personal agency. It includes 3 studies (plus sub studies) that differ in terms of their methodological approach as well as in the operationalization of the construct "personal agency" (i.e., sense of power, number of followers, depression). There is much to like about this research. I like its multi-method approach and the research question is relevant and interesting. This work extends past work by establishing this link between linguistic and personal agency empirically. Moreover, some of the studies were pre-registered.

I also have some concerns, which could be addressed in a revision.

1) The introduction is a bit superficial. For instance, I would like to see it better embedded in past research on power/agency and linguistics. The following papers may be relevant.

Wakslak, C., Smith, P. K., & Han, A., (2014). Using abstract language signals power. *JSPS*
Stel, M., van Dijk, E., Smith, P. K., van Dijk, W. W. & Djalal, F. M. (2012). Lowering pitch of your voice makes you feel more powerful and think more abstractly. *SPPS*

Puts, D. A., Hodges, C. R., Cardenas, R. A., & Gaulin, S.J.C., (2007). Men's voices as dominance signals: Vocal fundamental and format frequencies influence dominance attributions among men.

2) Personal agency is operationalized very differently in the three studies. I am wondering to what extent one may really be sure that all effects were driven by personal agency. Depression is about much more than just about a lack of agency. The number of followers may also reflect other aspects than simply agency. While I do find it meaningful and interesting to bundle these studies together in order to get a comprehensive picture, I find the narrative a bit problematic. It is unclear whether personal agency as a well-defined construct itself drove the effects. To solve this, it may help to present the predictors as different factors that have in common that they are characterized by greater personal agency (e.g., sense of power, number of followers, depression). In the discussion, it would be helpful to add a paragraph that critically discusses that personal agency may be a shared characteristic of all predictors but it is possible that the reported effects are not all exclusively driven by personal agency.

3) On page 5, it is described that Study 1 investigates "how lack of control over one's actions is reflected in passive language use". This suggest that there is a comparison of a low-power group with a control group. However, they contrast a high-power with a low-power group. This should be corrected. The study does not say anything about whether the effects were driven by high agency or a lack of agency.

4) Could it be that the power manipulation in Study 1 literally asked for differences in linguistic agency along with personal agency? In the high-power condition, one describes a situation in which one has power over others, thus one is the actor. In the low-power condition, participants are instructed to think about a situation in which someone else had power over them. So it seems like the other person is defined as the actor (i.e., grammatically, the other person is the subject of the sentence)? If this could indeed be an issue, maybe think of re-running the study in which you make sure there is no such confound in the instructions.

5) Page 13: Is there a typo in the year range? "data were posted between July 2019 and November 2016"

6) Given that there are often associations between masculinity and agency, I wonder whether there were any gender differences in your studies?

Dear Editor,

We wish to thank you and the reviewers for providing thoughtful comments and suggestions. We appreciate the time and effort put into reviewing our work. We have carefully considered the concerns raised in the reviews and have made the necessary revisions accordingly. Below, we detail the changes we have made. We included the reviewers' comments and our response in *italicized Arial*.

Reviewer #1:

[R1.1] This interesting paper connects the use of non-agentive language to social and personal agency. This is a well-written work describing three well-designed studies. I really like the paper and look forward to seeing it in print. Below, are a few suggestions that the author can consider in revising their work.

[Response R1.1] We thank the reviewer for his insightful suggestions and comments.

[R1.2] In all studies, the authors use a top-down measure of agentive language. Theoretically, this makes a lot of sense. But there could be interesting bottom-up or data-driven patterns as well, which could potentially shed light on previously unknown linguistic signatures of personal agency. The authors can conduct a simple topic modeling (e.g., LDA; see Maier et al., 2018) to examine what topics (or "themes") high-agency (vs. low-agency) people talk about in Studies 2 and 3 (on Twitter and Reddit).

[Response R1.2] Indeed, using bottom-up techniques could shed some light – mainly on the content rather than the style. In the tables below, we provide LDA topic modeling for each study. However, it is also known that LDA can produce results that are hard to interpret. As can be seen in the tables below, we think this is the case with our current data, and therefore, as it stands, we believe that these analyses may not inform the readers.

LDA Study 1

	low power					high power				
topic1	topic2	topic3	topic4	topic5	topic1	topic2	topic3	topic4	topic5	
questions	am	new	money	position	students	team	manage	some	his	
good	you	our	needed	supervisor	responsibility	group	off	want	after	
knew	their	day	said	person	however	working	worked	can	more	

school	i'm	years	some	manager	things	best	employe e	am	neede d
answer	can	year	allowed	working	whether	having	boss	don't	help

LDA Study 2:

less than 100 followers					more than 5000 followers				
topic1	topic2	topic3	topic4	topic5	topic1	topic2	topic3	topic4	topic5
was	are	that	on	my	be	at	it	are	your
that	@realdonal dtrump	do	at	me	what	with	my	that	this
my	not	be	with	i'm	it	this	was	his	my
he	they	if	your	so	that	from	that	has	we
been	that	what	day	on	they	are	just	an	our

LDA Study 3:

depression					control				
topic1	topic2	topic3	topic4	topic5	topic1	topic2	topic3	topic4	topic5
talking	die	mental	him	im	be	at	it	are	your
that's	wish	health	his	dont	what	with	my	that	this
used	pain	working	mom	its	it	this	was	his	my
social	hope	taking	said	stop	that	from	that	has	we
many	living	therapy	dad	sleep	they	are	just	an	our

[R1.3] In the top-down analysis of agentic language, the authors only use one kind of analysis. It would be interesting to conduct additional robustness checks using different top-down methods. For example, the authors can use the Contextualized Construct Representation method (CCR; Atari et al., 2023) - relying on a validated psychometric measure of agency, power, locus of control - to quantify agentic language. It is generally a good idea to use multiple methods when examining the relationship between language use

and psychological ground truth (for a discussion, see Kennedy et al., 2021).

[Response R1.3] We thank the reviewer for his valuable suggestions. In response, we have incorporated CCR analyses into all three studies, which reveal consistent evidence supporting the association between the conditions throughout the paper (low power, low social rank, depression) and constructs of depression (embedded CESD; Eaton et al., 2004), constrained sense of control (embedded SoC; Lachman & Weaver, 1998), and locus of control (embedded LoC; Rotter, 1966). The findings obtained from the CCR analyses closely align with the passive voice index.

However, in Study 2, while all effects were observed in the expected direction, our bootstrapped robust regression analysis indicates that the exponent of the coefficients (IRR) falls outside the boundaries of the 95% confidence interval. Nevertheless, when considering a more lenient confidence level, analogous to a one-sided test, the effects are evident in two out of the three constructs.

[R1.4] I think one important limitation of the current work is that it is based on one culture (i.e., American) and one language (i.e., English; see Blasi et al., 2023). This is fine, and the authors do touch on this toward the end of their Discussion, but in my opinion, it should be more explicitly voiced in the paper. Can it be the case that these findings are WEIRD (Henrich et al., 2010)? Do the authors expect to see the same pattern in non-English languages?

[Response R1.4] We appreciate the reviewer's suggestion to address this limitation more explicitly in our paper on p.22:

"One limitation of the current research is that it examines the relation between linguistic and personal agency only in a single cultural-linguistic context. While our current study does not provide direct evidence for this, we hypothesize that the relationship between personal agency and linguistic agency could potentially be observed in other languages as well, considering the universal nature of the concept of agency. However, the specific linguistic features that signal agency might differ across languages due to grammatical and cultural variations (Markus & Kitayama, 2003)."

[R1.5] Minor points:

I am really glad that the authors have provided annotations and some benchmarking. The Validation sections are rather short, though. It would be great if the authors could add more details to the Validation sections.

[Response R1.5] Thank you for drawing our attention to this point. We have now expanded the annotation details, which now include the following on p.6:

*“To validate that computational extraction of passive voice is indeed associated with non-agentive language, an independent rater coded 100 texts from this sample. The rater was presented with various texts, and their objective was to identify and count instances of non-agentive language within those texts. It was explained to the rater that non-agentive language refers to sentences where the action is described without specifying the person or entity responsible for it. On the other hand, agentive language includes sentences that clearly state the agent or individual who performed the action. In order to determine the number of non-agentive instances in each text, the rater needed to analyze the sentences and identify those that lack a specified agent. Prior to carrying out the rating procedure, the rater was provided with sufficient examples (e.g., *The vase broke* vs. *John broke the vase*; *The book was put on the table* vs. *Michelle put the book on the table*; *The curtain caught fire* vs. *I set the curtain on fire*). For the coding manual, see *Supplementary Materials*.”*

[R1.6] Please rephrase terms like “using big data”, “big data analyses”, and “big-data methods” as the term “big data” is vague with no added value. Instead, describe the exact nature of the method used (e.g., using terms like language analysis, text analysis, NLP, etc.).

[Response R1.6] We have now removed all instances of the term big data from the manuscript.

References

- Atari, M., Omrani, A., & Dehghani, M. (2023, February 24). Contextualized Construct Representation: Leveraging Psychometric Scales to Advance Theory-Driven Text Analysis. <https://doi.org/10.31234/osf.io/m93pd>
- Blasi, D. E., Henrich, J., Adamou, E., Kemmerer, D., & Majid, A. (2023). Over-reliance on English hinders cognitive science. *Trends in cognitive sciences*.
- Henrich, J., Heine, S. J., & Norenzayan, A. (2010). The weirdest people in the world?. *Behavioral and brain sciences*, 33(2-3), 61-83.
- Kennedy, B., Atari, M., Davani, A. M., Hoover, J., Omrani, A., Graham, J., & Dehghani, M. (2021). Moral concerns are differentially observable in language. *Cognition*, 212, 104696.
- Maier, D., Waldherr, A., Miltner, P., Wiedemann, G., Niekler, A., Keinert, A., ... & Adam, S. (2018). Applying LDA topic modeling in

communication research: Toward a valid and reliable methodology.
Communication Methods and Measures, 12(2-3), 93-118.

Signed review,
Mohammad Atari

Reviewer #2:

[R2.1] This review is for the manuscript "On Pulling the Strings or Having Your Strings Pulled: The Relation Between Personal and Linguistic Agency." Overall, there are many things to like about this manuscript: it is well-written, direct, and it is clear that the research was conducted thoughtfully. As I was reading this manuscript, two primary thoughts came to mind that I feel would help to strengthen it in terms of its appeal to a broader audience, as well as help to disambiguate some of the constructs that may be in play.

Major points of feedback:

One of the big things that I think would strengthen the manuscript would be to connect with a broader set of literatures that, historically, have been quite interested in understanding the psychology that underpins the use of active/passive voice, as well as other studies that show the consequences of framing thought in this fashion.

The authors note in the manuscript that a majority of research on active/passive voice and agency deals with the constructs of person perception/attribution. I absolutely agree with this sentiment, although, I might suggest that this is a slightly narrow view that is at the intersection of the fields of social psychology and psycholinguistics. Historically – at least through much of the 20th century if not before – psychotherapists, clinical psychologists, and discourse analysts have focused considerable attention on the relationship between linguistic agency and personal agency. Much of the work that I'm familiar with in these spaces is, I think, rooted largely in assumptions about the correspondence between the two, often deriving evidence for their relationship from clinical case studies and smaller samples. While none of this detracts from the current work in and of itself, I might suggest that the authors broaden the scope of the introduction/discussion sections leverage

the bodies of scholarship, practice, and academic thought in this area. I'll provide some references at the end of this review that may (or may not!) serve as helpful guideposts to some of the literature that I'm describing here. Note that I'm not suggesting that the authors need to cite all, or even any, of the references that I provide – rather, I might instead recommend them as examples that highlight a tradition of acknowledging the importance of the common ground between linguistic agency and personal agency, often in terms of understanding pretty central topics such as personal behavior, psychosocial change, mental health, emotion, and relationships. I'll also note that I'm passingly aware of a literature in criminology/justice/law that dovetails with some of the ideas raised in the existing literature, however, I'll admit that I'm far less familiar with work in these areas and, additionally, their relevance to the current research may not be as strong.

To wrap up my thought here: given the importance that is often placed on active-vs-passive voice in therapeutic contexts, connecting with this and other broader/richer traditions might expand the scope of the current work and, additionally, facilitate greater impact within several major areas of research and real-world outcomes.

[Response R2.1] Thank you for the opportunity to expand our historical context. We have now added the following to the introduction (p. 3):

“Previous psycholinguistic findings showed that linguistic framing influences the level of agency attributed to other people. For example, describing incidents in an agentive language led to greater attribution of blame by observers, and resulted in harsher punishments than in non-agentive language (Fausey & Boroditsky, 2010). Moreover, in languages that highlight the agent in accidental incidents (e.g., English), the agents are better encoded in memory than in languages that do not (e.g., Japanese) (Fausey et al., 2010). Such findings concerning the effects of agentive language join a rich body of work concerning linguistic (e.g., linguistic abstractness; Wakslak et al., 2014) and paralinguistic (e.g., voice pitch; Puts et al., 2007) cues that affect our attributions of personal agency to a third party. However, the question of how linguistic agency relates to the speaker’s own sense of agency has not been extensively studied.

Much of the previous work concerning the relation between linguistic and personal agency relied on qualitative discourse analyses. For example, a qualitative report suggests that individuals dealing with chronic pain often discuss their struggles using passive voice, supposedly reflecting a sense of reduced personal agency (Vroman et al., 2009). Furthermore, qualitative descriptions of people’s reconstructions of psychological therapy show that patients describe periods of psychological hardship in a passive voice and that they often use more agentive language when describing the process of improvement (Adler & McAdams, 2007).

To the best of our knowledge, two studies have conducted quantitative analyses of passive language use in the context of emotional hardships that may relate to a diminished sense of personal agency. Oren et al. (2016) have shown that individuals that suffer from OCD tend to use less agentive language. Additionally, Hughes et al. (1994) have shown that skin-conductance levels tend to increase when individuals describe traumatic experiences in a passive voice. This finding suggests that passive language was associated with experiences of high negative arousal, during the recollection of events that are often characterized by deprivation of control”.

[R2.2] Related to the above, one big set of concepts that might be meaningful to delve into and disambiguate with the current set of studies are those of psychological distance, immediacy, and internal-vs-external focus. I'm treating all of these as fairly unified, conceptually speaking (although I know that there are nuances in how they are studied, understood and described), in part, because I believe that the heart of a lot of these and related constructs generally relate to a focus on oneself versus the environment (or, sometimes, the relationship/distance between the two). There is a decent amount of work that suggests that active-vs-passive framing is primarily (or at least, in part) a reflection of whether a person is focusing on oneself or something in the environment (e.g., another person, an object in the environment, etc.). For example, some work shows that active-vs-passive framing is related to goal attainment (e.g., focusing on one's own hunger vs. the qualities of the food being consumed in a weight loss context; "I ate the cake" versus "the cake was eaten by me"). It might be possible, then, to characterize or consider at least some of the current findings through a lense of "self versus external." For example, Study 1's results may not only reflect a sense of personal agency, but might also be a function of prompting a person's focus on their power versus the power of other individuals.

A potentially interesting way to help disambiguate personal agency from self/other would be to do an additional check on the use of first-person singular pronouns as subject (I, I'm, I'll, etc.) versus object (e.g., "me"). Do high and low linguistic agency correspond to self-as-subject and self-as-object, respectively? I don't have a sense of how this might work in the context of the Twitter data, as many of the tweets may not involve the self at all (e.g., "This website is having a sale on socks!"), but my intuition is that the self-narratives in Study 1 and the r/depression subreddit posts would be likely to invoke the self quite a bit. Or, alternatively (and perhaps more simply) – is there an inverse relationship between passive voice and self-referential language? My sense is that,

regardless of the results, this might be an interesting thing to report that would also help to contextualize the current findings and lend to a more rich and informative interpretation of what is happening, psychologically speaking, when a person uses the passive voice.

[Response R2.2] We thank the reviewer for his thoughtful comment. Considering the self as an object or subject is indeed an interesting observation and may very well be worthy of further research. We agree that self-referential language (whether object or subject) is an important factor reflecting agency, and we have now added it as a factor and dependent variable to all our models throughout the manuscript. We elaborate on the rationale of self-referential language on p.5:

“Another manifestation of agentive language, is the use of self-referential language. The perception of self-agency entails that the self is the causer of events (e.g., “I call the shots”); as such, an additional, potentially important, dimension of agentive language is how frequently individuals refer to themselves in their narratives (Adler & McAdams, 2007). While such self-referential language may be a marker of self-agency, much previous research has shown that self-referential language is increased during depressive episodes, supposedly due to the increased self-focus that is common in depression (e.g., Rude et al., 2004); because depression is related to reduced self-agency (see discussion in Study 3), it is also possible that self-referential processing will actually be a correlate of reduced self-agency. Given these competing possibilities, we did not have a directional hypothesis concerning the effect of self-referential language and included it in our analysis for exploratory purposes”

We further expand in the Discussion section on p.21:

“While our main analysis pertained to the use of the passive voice, we also conducted an exploratory investigation of how personal agency is related to people’s use of self-referential language. On one hand, the perception of self-agency entails that the self is the causer of events; as such, one might expect that I-language will be correlated with higher personal agency. On the other hand, self-reference is a known correlate of depression, which is characterized by a markedly reduced sense of self-agency, and as such may be associated with lower personal agency. Across all three studies, lower personal agency (i.e., lack of control, lower social rank, depression) was associated with greater use of self-referential language. These findings replicate the well-known relation between personal pronoun use and depression (Rude et al., 2004), and highlight that this relation generalizes to broader lower agency states (e.g., lower social rank, lack of control).”

In addition, we believe that certain concerns raised about the emphasis on self versus external factors have been effectively addressed through the CCR analyses recommended by Reviewer 1, specifically through the inclusion of the Locus of Control construct.

[R2.3]Smaller thoughts:

One thing that I was curious about was the nature of the data for both Twitter and Reddit: is it the case that all of the texts were "top level" posts? For example, were all of the Reddit posts analyzed "submissions" rather than comments in response to submissions? Likewise, did the tweet corpus include both initial tweets as well as responses/comments to other tweets? Going back to the idea of self-versus-other focus, I can imagine a bit of difficulty in separating the idea of "I'm tweeting in response to another person" and the use of passive language, which might reflect "I'm focusing on you" (whereas a top-level tweet might be more likely to be about oneself, and use less passive voice). It might also be the case that people who have more followers would be more likely to make, rather than respond to, other tweets.

I don't think that any of those ideas need to be fleshed out in the paper itself, but it might be helpful to provide a bit of additional information (in supplementary materials, perhaps?) about the data itself. And, of course, if this information is already present and I missed it, please accept my apologies!

[Response R2.3] Thank you for your comment. In our study, all Reddit posts were considered as top-level "submissions" rather than "comments." However, the Twitter data we collected included top- and lower-level posts, which encompassed posts and replies. To examine the relationship between passive voice usage and message level, we used the same random sample from the CCR analyses (R1.3) with a total of $N = 100,000$ posts. After excluding multiple posts by individual users, the sample size was reduced to $N = 81,606$. In our analysis, we investigated how the use of passive voice varied based on the message level (i.e., top-level vs. replies), controlling for tweet length. We found that the main effect of passive voice remained significant and exhibited the same direction ($IRR = 0.82, p < .001$). Additionally, we observed a main effect of message level, indicating that replies were associated with users who had fewer followers ($IRR = 0.50, p < .001$). We did not find evidence of an interaction between passive voice usage and message level ($IRR = 0.93, p = 0.17$). We have now included this analysis in the Supplementary Materials.

[R2.4] Again, a very minor thought: I would be curious to know whether submissions to the various subreddits receive more comments or upvotes if they use less passive language, which might help reinforce the findings from Study 2. Again, nothing that I would necessarily include in the manuscript itself unless the authors feel that it strengthens the paper, and this is definitely not a major thing that I think even needs to be explored if the authors feel that it's too tangential.

[Response R2.4]. The main focus of Study 2 is to establish a connection between social power and the use of passive voice. On Twitter, social power is evident through the number of followers an individual has. However, on Reddit, although there may be indicators of social power, they are not as overt or easily quantifiable.

It's important to note that engagement is a distinct factor that is associated with social power, but it primarily reflects the virality or popularity of posts rather than the power held by the author. Therefore, we believe that conducting an engagement analysis would not provide clear insights for our prediction or further contribute to our understanding of the relationship between social power and the use of passive voice.

Nevertheless, as we do understand why the results may be interesting, we report them here:

We report an engagement analysis of Study 3b (as in Study 3a the model suffered from technical issues). We define engagement as the number of comments for the top-level post. We find that number of comments is positively associated with passive voice (though, note the small effect size: $IRR = 1.0002$, $p = .029$); the depression group was still positively associated with greater passive voice ($IRR = 1.40$, $p < .001$). We also find an interaction, such that in the depression subreddit, we observed that posts with a higher number of comments were associated with reduced use of passive voice ($IRR = 0.996$, $p = .039$). See figure:

Supporting Figure 1. Interaction between number of comments (showing interquartile range) and group. Values on the Y-axis denote predicted passive voice count. Error bars denote standard errors.

[R2.5]For the depression subreddit findings, one concern that I have is that the depression subreddit is fairly unique from most (perhaps all?) of the comparison subreddits in that it is generally framed as a place for peer support & help-seeking. While the assumption is fair that the experience of depression is the major contributor to the findings here, I might suggest a more analogous set of comparison groups than a random selection of subreddits. I do know that it can be difficult to find support-seeking groups where there is a low likelihood of comorbidity with depression (for example, people suffering from medical conditions are likely also going to be under emotional duress), there may be several subreddits that focus on support/help-seeking that are less affectively tinged (e.g., r/cartalk is a subreddit for help with car trouble, r/personalfinance

for people looking for financial help and advice, r/fashionadvice for people looking for fashion feedback, etc.).

[Response R2.5] Thank you for the great advice. We have now added Study 3c, in which we provide a comprehensive set of control subreddits devoted to support and advice sharing that do not share comorbidity with depression (e.g., cooking advice, medical advice, programming support, etc.). We replicate our main finding, showing greater passive voice in the depression subreddit vs. the new control (IRR = 1.05, $p < .001$). See more details on p.17.

[R2.6] Lastly, and another very minor note: the current title of "On Pulling the Strings or Having Your Strings Pulled" rang a bit empty for me in terms of signalling the studies in the manuscript. Initially, it read to me more along the lines of classic findings in the way of person perception/attribution studies – is a person responsible for their actions, or was it some external force? Of course, the phrasing is relevant to the active/passive distinction, but I might suggest something that more clearly relates to whether a person is high/low in personal agency rather than something that might be interpreted as an attributional cause of one's actions.

[Response R2.6] We have now changed the title to A Computational Text Analysis Investigation of the Relation Between Personal and Linguistic Agency.

In sum, I want to re-emphasize that I found this to be a very nice and interesting manuscript, and I hope that the authors find this feedback helpful!

Sincerely,
Ryan L. Boyd

Additional references:

Vroman, K., Warner, R., & Chamberlain, K. (2009). Now let me tell you in my own words: Narratives of acute and chronic low back pain. *Disability and Rehabilitation*, 31(12), 976–987.
<https://doi.org/10.1080/09638280802378017>

Adler, J. M., & McAdams, D. P. (2007). The narrative reconstruction of psychotherapy. *Narrative Inquiry*, 17(2), 179–202.
<https://doi.org/10.1075/ni.17.2.03adl>

Quayle, E., & Taylor, M. (2002). Child pornography and the internet: Perpetuating a cycle of abuse. *Deviant Behavior*, 23(4), 331-361.
<https://doi.org/10.1080/01639620290086413>

Szalai K., & László J. (2007). Az aktív és passzív igék gyakorisága a csoportjelenségek tükrében: Történelemkönyvek szövegeinek narratív pszichológiai vizsgálata NooJ tartalomelemző programmal [Measuring the frequency of active and passive verbs in historical narratives: Narrative psychological analysis of the texts of history books with the NooJ program]. *Magyar Számítógépes Nyelvészeti Konferencia*, 5, 242-249. <http://acta.bibl.u-szeged.hu/58682/>

László, J., Csertő, I., Fülöp, É., Ferenczhalmy, R., Hargitai, R., Lendvai, P., Péley, B., Pólya, T., Szalai, K., Vincze, O., & Ehmann, B. (2013). Narrative language as an expression of individual and group identity: The narrative categorical content analysis. *SAGE Open*, 3(2), 2158244013492084.
<https://doi.org/10.1177/2158244013492084>

Senay, I., Usak, M., & Prokop, P. (2015). Talking about behaviors in the passive voice increases task performance. *Applied Cognitive Psychology*, 29(2), 262-270. <https://doi.org/10.1002/acp.3104>

Demetci, P., Nichols, C., Zastavker, Y. V., Stolk, J. D., Dillon, A., & Gross, M. D. (2016). Internalization and externalization in the classroom: How do they emerge and why is it important? 2016 IEEE Frontiers in Education Conference (FIE), 1-5.
<https://doi.org/10.1109/FIE.2016.7757463>

Schwartzman, R., & Boger, K. E. (2019). How international students using communication centers navigate locus of control. In A. G. Mag (Ed.), *Perspectives of Arts and Social Studies: Vol. Vol. 3* (pp. 36-48). Book Publisher International.
<https://doi.org/10.9734/bpi/pass/v3>

Hodge, C., Pederson, J. A., & Walker, M. (2015). How do you "like" my style? Examining how communication style influences Facebook behaviors. *International Journal of Sport Communication*, 8(3), 276-292. <https://doi.org/10.1123/IJSC.2015-0052>

Kameny, R. R., & Bearison, D. J. (2002). Cancer narratives of adolescents and young adults: A quantitative and qualitative analysis. *Children's Health Care*, 31(2), 143-173.
https://doi.org/10.1207/S15326888CHC3102_5

Ferreira, F. (1994). Choice of passive voice is affected by verb type and animacy. *Journal of Memory and Language*, 33(6), 715-736.
<https://doi.org/10.1006/jmla.1994.1034>

Van Staden, C. W., & Fulford, K. W. M. (2004). Changes in semantic uses of first person pronouns as possible linguistic markers of recovery in psychotherapy. *The Australian and New Zealand Journal of Psychiatry*, 38(4), 226-232.
<https://doi.org/10.1080/j.1440-1614.2004.01339.x>

Manthei, R. J., & Matthews, D. A. (1982). Helping the reluctant client to engage in counselling. *British Journal of Guidance & Counselling*, 10(1), 44-50. <https://doi.org/10.1080/03069888208258038>

Gupta, A. (2018). The effect of mindfulness, cognitive behavior therapy and academic training on conduct and academic problems of disadvantaged children. *Journal of Psychology & Psychotherapy*, 8(4), 1-12. <https://doi.org/10.4172/2161-0487.1000347>

Hughes, C. F., Uhlmann, C., & Pennebaker, J. W. (1994). The body's response to processing emotional trauma: Linking verbal text with autonomic activity. *Journal of Personality*, 62(4), 565-585.
<https://doi.org/10.1111/j.1467-6494.1994.tb00309.x>

Brady, J. P. (2018). Language in schizophrenia. *American Journal of Psychotherapy*.
<https://doi.org/10.1176/appi.psychotherapy.1958.12.3.473>

Lorenz, M. (1953). Language as expressive behavior. *A.M.A. Archives of Neurology & Psychiatry*, 70(3), 277-285.
<https://doi.org/10.1001/archneurpsyc.1953.02320330002001>

Lorenz, M. (1955). Expressive behavior and language patterns. *Psychiatry*, 18(4), 353-366.
<https://doi.org/10.1080/00332747.1955.11023020>

Reviewer #3:

[R3.1] The present paper investigates whether there is a link between linguistic agency and personal agency. It includes 3 studies (plus sub studies) that differ in terms of their methodological approach as well as in the operationalization of the construct "personal agency" (i.e., sense of power, number of followers, depression). There is

much to like about this research. I like its multi-method approach and the research question is relevant and interesting. This work extends past work by establishing this link between linguistic and personal agency empirically. Moreover, some of the studies were pre-registered.

I also have some concerns, which could be addressed in a revision.

1) The introduction is a bit superficial. For instance, I would like to see it better embedded in past research on power/agency and linguistics. The following papers may be relevant.

Wakslak, C., Smith, P. K., & Han, A., (2014). Using abstract language signals power. JPSP

Stel, M., van Dijk, E., Smith, P. K., van Dijk, W. W. & Djalal, F. M. (2012). Lowering pitch of your voice makes you feel more powerful and think more abstractly. SPPS

Puts, D. A., Hodges, C. R., Cardenas, R. A., & Gaulin, S.J.C., (2007). Men's voices as dominance signals: Vocal fundamental and formant frequencies influence dominance attributions among men.

[Response R3.1] Thank you for the reference. We have now added the following on p.3:

“Previous psycholinguistic findings showed that linguistic framing influences the level of agency attributed to other people. For example, describing incidents in an agentive language led to greater attribution of blame by observers, and resulted in harsher punishments than in non-agentive language (Fausey & Boroditsky, 2010). Moreover, in languages that highlight the agent in accidental incidents (e.g., English), the agents are better encoded in memory than in languages that do not (e.g., Japanese) (Fausey et al., 2010). Such findings concerning the effects of agentive language join a rich body of work concerning linguistic (e.g., linguistic abstractness; Wakslak et al., 2014) and paralinguistic (e.g., voice pitch; Puts et al., 2007) cues that affect our attributions of personal agency to a third party. However, the question of how linguistic agency relates to the speaker's own sense of agency has not been extensively studied.

Much of the previous work concerning the relation between linguistic and personal agency relied on qualitative discourse analyses. For example, a qualitative report suggests that individuals dealing with chronic pain often discuss their struggles using passive voice, supposedly reflecting a sense of reduced personal agency (Vroman et al., 2009). Furthermore, qualitative descriptions of people's reconstructions of psychological therapy show that patients describe periods of psychological hardship in a passive voice and that they often use more agentive language when describing the process of improvement (Adler & McAdams, 2007).

To the best of our knowledge, two studies have conducted quantitative analyses of passive language use in the context of emotional hardships that may relate to a diminished sense of personal agency. Oren et al. (2016) have shown that individuals that suffer from OCD

tend to use less agentive language. Additionally, Hughes et al. (1994) have shown that skin-conductance levels tend to increase when individuals describe traumatic experiences in a passive voice. This finding suggests that passive language was associated with experiences of high negative arousal, during the recollection of events that are often characterized by deprivation of control”.

[R3.2] Personal agency is operationalized very differently in the three studies. I am wondering to what extent one may really be sure that all effects were driven by personal agency. Depression is about much more than just about a lack of agency. The number of followers may also reflect other aspects than simply agency. While I do find it meaningful and interesting to bundle these studies together in order to get a comprehensive picture, I find the narrative a bit problematic. It is unclear whether personal agency as a well-defined construct itself drove the effects. To solve this, it may help to present the predictors as different factors that have in common that they are characterized by greater personal agency (e.g., sense of power, number of followers, depression). In the discussion, it would be helpful to add a paragraph that critically discusses that personal agency may be a shared characteristic of all predictors but it is possible that the reported effects are not all exclusively driven by personal agency.

[Response R3.2] We agree with the reviewer and now have added clarification in the introduction (p. 4):

“ Specifically, we aimed to explore whether various factors characterized by personal agency are reflected in the extent to which individuals use the passive voice.”

Second, we now added the CCR analysis that provides some evidence to the claim that across all studies, there is a shared component of lower personal agency (i.e., constrained sense of control; external locus of control; higher depression).

Moreover, we now acknowledge and elaborate on this approach in the discussion section. We note that sense of power, depression and number of followers may indeed encompass other aspects and characteristics beyond personal agency. Specifically, we emphasized that personal agency may be a shared characteristic of all predictors, but other factors may also contribute to the observed effects. See p.21-22:

“It is important to note that while our findings suggest a relationship between personal agency and linguistic agency, the reported effects may not be exclusively driven by personal agency. The predictors examined in our study, such as sense of power, number of followers, and depression, share the common characteristic of personal agency (Calhoun et al., 1974; Caspar

et al., 2018; Rapee et al., 1996; Schwarz et al., 2023; Seligman, 1973). However, other factors and aspects related to these predictors may also contribute to the observed effects on linguistic agency. For instance, depression encompasses a wide range of symptoms and experiences beyond a lack of agency, and the number of followers may reflect various aspects of an individual's online presence, such as social status or content quality. Similarly, the sense of power can be influenced by numerous personal and situational factors that may affect linguistic agency independently. Thus, while personal agency is a shared characteristic of all predictors, we acknowledge the possibility that the reported effects are not all exclusively driven by personal agency. In other words, when considering each study individually, each one focuses on specific aspects of the complex connection between language and personal agency. However, when we examine these studies collectively, we believe that they contribute to our understanding of how language reflects individuals' sense of personal agency."

[R3.3] On page 5, it is described that Study 1 investigates "how lack of control over one's actions is reflected in passive language use". This suggest that there is a comparison of a low-power group with a control group. However, they contrast a high-power with a low-power group. This should be corrected. The study does not say anything about whether the effects were driven by high agency or a lack of agency.

[Response R3.3] Thank you for the correction. We now explicitly mention that it was contrasted with high power over others. P.4:

"In Study 1, we examined how participants' lack of sense of agency (i.e., little power over one's actions vs. high power) is reflected in passive language use"

[R3.4] Could it be that the power manipulation in Study 1 literally asked for differences in linguistic agency along with personal agency? In the high-power condition, one describes a situation in which one has power over others, thus one is the actor. In the low-power condition, participants are instructed to think about a situation in which someone else had power over them. So it seems like the other person is defined as the actor (i.e., grammatically, the other person is the subject of the sentence)? If this could indeed be an issue, maybe think of re-running the study in which you make sure there is no such confound in the instructions.

[Response R3.4] The reviewer highlights a valid concern: the low-power condition, wherein someone has power over the self, may increase the probability that the narrator is often the object, rather than the subject of the sentence. If this is the case, it is then possible that sentences where the narrator is the object (vs. subject) are more likely to be written in the passive voice.

We agree that this is a valid concern, however, our new analyses suggest that this possibility does not apply to our data.

First, it is important to note that the passive\active distinction is, at least, in principle independent from the subject\object dimension. Most passive sentences appear with the narrator as the subject of the sentence (“I was pushed around by Dan”), and sentences where the narrator is the object are often active (“Dan pushed me around”). Importantly, in the revised manuscript we also analyzed the extent to which the texts include self-referential language (e.g., I, Me). We saw that the effect of social power on passive language is independent of the use of such self-referential language, and that the effect exists also when no self-referential language is used. For a sentence to refer to the self in the object or subject position, it must contain a personal pronoun (e.g., I was pushed around; Dan pushed me). However, in the new analyses, we saw that there is an increase in passive language in the low power position also when the self is neither in the subject or object position (e.g., the iphone got broken), as seen in the figure below. These findings indicate that the effect of power on language is not the result of increased reference to the self as an object.

Supporting Figure 2. Study 1 effect of group on passive auxiliary verbs, imposing I-words = 0.

[R3.5] Page 13: Is there a typo in the year range? “data were posted between July 2019 and November 2016”

[Response R3.5] Thank you for the correction. We now fixed the range to November 2016 and July 2019. If the comment was about the wide range, then the reason for this significant expansion is insufficient data collected from the API. As a result, the search algorithm has made an automatic decision to broaden the search range .

[R3.6] Given that there are often associations between masculinity and agency, I wonder whether there were any gender differences in your studies?

[Response R3.6] Thank you for the suggestion. The only dataset where we have the ability to perform such analysis is Study 1, as gender is not overtly reported on social media. Following your suggestion we added gender to the modeling of Study 1. Descriptively, in the High power condition, we find that women use the passive voice to a greater extent (IRR = 1.03); however, that is not a significant increase ($p = 0.781$). In the Low Power condition, women tend to use passive voice to a lesser extent (IRR = 0.98); however, this result is not significant either ($p = .371$). We now report these results in Table S2 in the Supplementary Materials.

27th Jul 23

Dear Dr Simchon,

Your manuscript titled "A Computational Text Analysis Investigation of the Relation Between Personal and Linguistic Agency" has now been seen by our reviewers, whose comments appear below. In light of their advice I am delighted to say that we are happy, in principle, to publish a suitably revised version in Communications Psychology under the open access CC BY license (Creative Commons Attribution v4.0 International License).

Reviewers #1 and #3 from the previous round are satisfied that all their concerns were addressed. As for the concerns voiced by Reviewer #2 previously, they felt that the literature treatment was still not satisfactorily, as a significant body of work remains neglected.

We therefore invite you to revise your paper one last time to address the remaining concerns of our reviewers and a list of editorial requests. At the same time we ask that you edit your manuscript to comply with our format requirements and to maximise the accessibility and therefore the impact of your work.

Related to the concerns previously voiced by Reviewer #2 about missing additional background literature, we editorially consider the Introduction to contain too many implicit novelty claims, where it would be more appropriate to point to a body of related work. In the Editorial Requests Table, we highlight which statements need to be removed to avoid unwarranted suggestions of novelty.

Moreover, we require you to revise the overall presentation, and in particular the Abstract to present a summary of the study that is much closer to the data. Your work is comprehensive and combines data from many sources, including replications of the analysis in study 3, which is a clear strength. At the same time, the analyses are either correlational or derived from a between-participants design (or both), and the measure/approximation of agency differs significantly between studies, especially Study 3. We appreciate the discussion of these issues as limitations, but to avoid a misreading or misrepresentation of your work, we also require you to amend the presentation in other places. I include an edited Abstract in the Editorial Requests Table.

EDITORIAL REQUESTS:

Please review our specific editorial comments and requests regarding your manuscript in the attached "Editorial Requests Table". Please outline your response to each request in the right hand column. Please upload the completed table with your manuscript files as a Related Manuscript file. We will not be able to proceed towards acceptance of your manuscript unless all editorially highlighted issues are entirely resolved.

SUBMISSION INFORMATION:

OPEN ACCESS:

Communications Psychology is a fully open access journal. Articles are made freely accessible on publication under a [CC BY](http://creativecommons.org/licenses/by/4.0) license (Creative Commons Attribution 4.0 International License). This license allows maximum

dissemination and re-use of open access materials and is preferred by many research funding bodies.

For further information about article processing charges, open access funding, and advice and support from Nature Research, please visit <https://www.nature.com/commspsychol/article-processing-charges>

At acceptance, you will be provided with instructions for completing this CC BY license on behalf of all authors. This grants us the necessary permissions to publish your paper. Additionally, you will be asked to declare that all required third party permissions have been obtained, and to provide billing information in order to pay the article-processing charge (APC).

* **DATA AVAILABILITY:**

[link redacted]

Best regards,

Antonia Eisenkoeck

Antonia Eisenkoeck
Senior Editor
Communications Psychology

REVIEWERS' COMMENTS:

Reviewer #1 (Remarks to the Author):

I thank the reviewers for carefully incorporating the suggestions into their work in this round of review. I have now had the opportunity to read the revised version and enjoyed it. As I mentioned to the Editors, the authors presented offer a novel and insightful investigation using cutting-edge computational text analysis methods. The authors employ a rigorous analytical approach and provide clear explanations of their methods and findings. I recommend publication.

Reviewer #3 (Remarks to the Author):

All my concerns have been addressed satisfactorily. I recommend this paper to be accepted for publication